# Conducting active screening for human African trypanosomiasis with rapid diagnostic tests: The Guinean experience (2016–2021)

**Oumou Camara[1], Justin Windingoudi Kaboré[1], Aïssata Soumah[1], Mamadou Leno[1], Mohamed Sam Bangoura[1], Dominique N'Diaye[2], Adrien Marie Gaston Belem[3], Sylvain Biéler[4], Mamadou Camara[1], Jean-Mathieu Bart[1,5], Brice Rotureau[2,6]*, Bruno Bucheton[1,5]**

**1** Programme National de Lutte contre la Trypanosomiase Humaine Africaine, Ministère de la Santé, Conakry, Guinea, **2** Parasitology Unit, Institut Pasteur of Guinea, Conakry, Guinea, **3** Institut du développement rural (IDR), Université polytechnique de Bobo-Dioulasso (UPB), Bobo-Dioulasso, Burkina Faso, **4** Foundation for Innovative New Diagnostics (FIND), Geneva, Switzerland, **5** Institut de Recherche pour le Développement, Unité Mixte de Recherche IRD-CIRAD 177 InterTryp, Campus International de Baillarguet, Montpellier, France, **6** Trypanosome Transmission Group, Trypanosome Cell Biology Unit, INSERM U1201, Institut Pasteur, Université de Paris, Paris, France

☯ These authors contributed equally to this work.
* rotureau@pasteur.fr

**Data Availability Statement:** The authors confirm that all data underlying the findings are fully available without restriction. All relevant data are

## Abstract

Strategies to detect Human African Trypanosomiasis (HAT) cases rely on serological screening of populations exposed to trypanosomes. In Guinea, mass medical screening surveys performed with the Card Agglutination Test for Trypanosomiasis have been progressively replaced by door-to-door approaches using Rapid Diagnostic Tests (RDTs) since 2016. However, RDTs availability represents a major concern and medical teams must often adapt, even in the absence of prior RDT performance evaluation. For the last 5 years, the Guinean HAT National Control Program had to combine three different RDTs according to their availability and price: the SD Bioline HAT (not available anymore), the HAT Sero-K-SeT (most expensive), and recently the Abbott Bioline HAT 2.0 (limited field evaluation). Here, we assess the performance of these RDTs, alone or in different combinations, through the analysis of both prospective and retrospective data. A parallel assessment showed a higher positivity rate of Abbott Bioline HAT 2.0 (6.0%, n = 2,250) as compared to HAT Sero-K-SeT (1.9%), with a combined positive predictive value (PPV) of 20.0%. However, an evaluation of Abbott Bioline HAT 2.0 alone revealed a low PPV of 3.9% (n = 6,930) which was surpassed when using Abbott Bioline HAT 2.0 in first line and HAT Sero-K-SeT as a secondary test before confirmation, with a combined PPV reaching 44.4%. A retrospective evaluation of all 3 RDTs was then conducted on 189 plasma samples from the HAT-NCP biobank, confirming the higher sensitivity (94.0% [85.6–97.7%]) and lower specificity (83.6% [76.0–89.1%]) of Abbott Bioline HAT 2.0 as compared to SD Bioline HAT (Se 64.2% [52.2–74.6%]—Sp 98.4% [94.2–99.5%]) and HAT Sero-K-SeT (Se 88.1% [78.2–93.8%]—Sp 98.4% [94.2–99.5%]). A comparison of Abbott Bioline HAT 2.0 and malaria-RDT positivity rates on 479 subjects living in HAT-free malaria-endemic areas further revealed that a significantly higher proportion of subjects positive in Abbott Bioline HAT 2.0 were also positive in

within the paper and its Supporting Information files.

**Funding:** This work was supported by the Bill & Melinda Gates Foundation (http://www.gatesfoundation.org, grants OPP1033712, OPP1154033 and INV-001785). JWK was funded by a TrypanoGEN AAS/Welcome post-doc grant. The funders had no role in study design, data collection and analysis, decision to publish, or preparation of the manuscript.

**Competing interests:** The authors have declared that no competing interests exist.

malaria-RDT, suggesting a possible cross-reaction of Abbott Bioline HAT 2.0 with malaria-related biological factors in about 10% of malaria cases. This would explain, at least in part, the limited specificity of Abbott Bioline HAT 2.0. Overall, Abbott Bioline HAT 2.0 seems suitable as first line RDT in combination with a second HAT RDT to prevent confirmatory lab overload and loss of suspects during referral for confirmation. A state-of-the-art prospective comparative study is further required for comparing all current and future HAT RDTs to propose an optimal combination of RDTs for door-to-door active screening.

## Author summary

Strategies to detect Human African Trypanosomiasis (HAT) cases rely on serological screening of populations exposed to trypanosomes. In Guinea, mass medical screening surveys performed with the Card Agglutination Test for Trypanosomiasis have been progressively replaced by door-to-door approaches using Rapid Diagnostic Tests (RDTs) since 2016. However, RDTs availability represents a major concern and medical teams must often adapt, even in the absence of prior RDT performance evaluation. For the last 5 years, the Guinean HAT National Control Program had to combine three different RDTs according to their availability and price: the SD Bioline HAT (not available anymore), the HAT Sero-K-SeT (most expensive), and recently the Abbott Bioline HAT 2.0 (limited field evaluation). Here, we assess the performance of these RDTs, alone or in different combinations, through the analysis of both prospective and retrospective data. Overall, Abbott Bioline HAT 2.0 seems suitable as first line RDT in combination with a second HAT RDT to prevent confirmatory lab overload and loss of suspects during referral for confirmation. A state-of-the-art prospective comparative study is further required for comparing all current and future HAT RDTs to propose an optimal combination of RDTs for door-to-door active screening.

## Introduction

Human African Trypanosomiasis (HAT) or sleeping sickness is a neglected tropical disease that occurs in sub-Saharan Africa, within the tsetse fly vector distribution limits [1]. HAT is caused by two trypanosome subspecies: in West and Central Africa, *Trypanosoma brucei (T. b.) gambiense* causes 87% of all reported HAT cases and is responsible for the chronic form of the disease (*gambiense* HAT); and the remaining 13% of HAT cases are due to *T. b. rhodesiense* in East Africa that leads to an acute infection [2].

In 2012, the World Health Organization (WHO) published a roadmap targeting *gambiense* HAT elimination as public health problem (PHP) as a main goal by 2020 [3]. In Guinea, this objective has been delayed due to the Ebola outbreak that impeded medical surveys from 2014 to 2016 [4]. Nevertheless, thanks to the substantial efforts of the Guinean HAT National Control Program (HAT-NCP), elimination of HAT as PHP is now about to be validated and *T. b. gambiense* transmission interruption is planned by 2030. To reach this objective, an efficient and continuous epidemiological surveillance will be key. Identification of HAT cases first rely on the immunodetection of specific antibodies in individuals with adapted serological tests, either by active screening or passive diagnostic. The Card Agglutination Test for Trypanosomiasis (CATT/*T. b. gambiense*) has been used for decades and still represents the gold standard serological method. This method perfectly fits the mass screening surveys, despite the

requirement of electric supply and cold chain for reagent conservation. New methods such as Rapid Diagnostic Tests (RDTs), which can be kept at ambient temperature, have been developed in the last decade. In Guinea, the first introduction of an RDT in 2014 was accompanied by the establishment of a surveillance network of 105 peripheral health centers in charge of passive diagnostic [4]. In the meantime, the use of RDTs has also become increasingly important for active screening with the development of door to door (D2D) screening campaigns. D2D active screening was first launched in Guinea after the Ebola outbreak (2013–2016) to regain population trust, and it progressively replaced mass screening campaign with CATT when the COVID-19 pandemic started. RDTs present significant advantages for D2D active screening: (i) they do not require neither a cold chain nor electricity, (ii) their simple handling only requires minimal training, hence skilled local health agents can be released for other medical tasks, (iii) their flexibility of use allows for more intimate diagnosis and limits population gathering, a strong argument during the COVID-19 pandemic.

Two RDTs have been commercialized and are available in the field: the HAT Sero-K-SeT supplied by Coris BioConcept (Belgium) and the Abbott Bioline HAT 2.0 supplied by Abbott (South Korea), that replaced the SD Bioline HAT in 2021 [5–7]. From 2016 to 2018, only SD Bioline HAT has been used in the three Guinean foci (Boffa, Dubreka and Forecariah) as presented in Camara et al, 2021 [4] (Fig 1). In 2019 and 2020, the uncertainty regarding SD Bioline HAT availability became a major concern and it was decided to use HAT Sero-K-SeT as the primary test for D2D screening, while maintaining the SD Bioline HAT available for passive surveillance (Fig 1). In 2021, Abbott Bioline HAT 2.0 became available, and its low price, as compared to HAT Sero-K-SeT, made this test the primary RDT for D2D (Fig 1). However, no comparative studies were available to assess its performance in the field.

The present study describes the two major challenges that the HAT-NCP had to face since 2016 to implement a cost-effective strategy for active screening with RDTs: (1) to minimize false positive results tending to overload confirmation laboratories, (2) to maintain a sufficient positive predictive value (PPV) in transmission foci where HAT incidence has been continuously decreasing. It is based on a combination of fragmented retrospective observations and prospective evaluations collected over 6 years of medical surveys on more than 46,000 individuals. It especially provides comparisons of the new Abbott Bioline HAT 2.0 to the previous RDTs used in Guinea.

## Material and methods

### Ethics statement

All the prospective work done in the field in the three HAT foci was performed during the medical surveillance campaigns conducted by the HAT National Control Program (HAT-NCP) of the Republic of Guinea that was established by the Ministry of Health in 2002. An open authorization for anonymous blood sampling and conservation, as well as analysis of diagnostic screening results for epidemiological surveillance was provided to the HAT-NCP by the Ministry of Health of Guinea in the framework of the ElimTryGui roadmap of the HAT-NCP. For the retrospective analyses of the biobanked sera and the prospective comparison of malaria and HAT RDTs, the Comité National Ethique en Recherche Scientifique (CNERS) of Guinea provided an agreement prior to the present study (Approval Diag-Cut-THA #032/CNERS/17 and amendment #038/CNERS/19). For the retrospective analyses, all sera originated from the HAT-NCP Biobank for which participants or their legal representatives signed a written informed consent before enrolment (Studies Diag-Cut-THA 032/CNERS/17 and amendment 038/CNERS/19). During all the study, after obtention of an individual signed informed consent, each participant was attributed a unique code / identifier

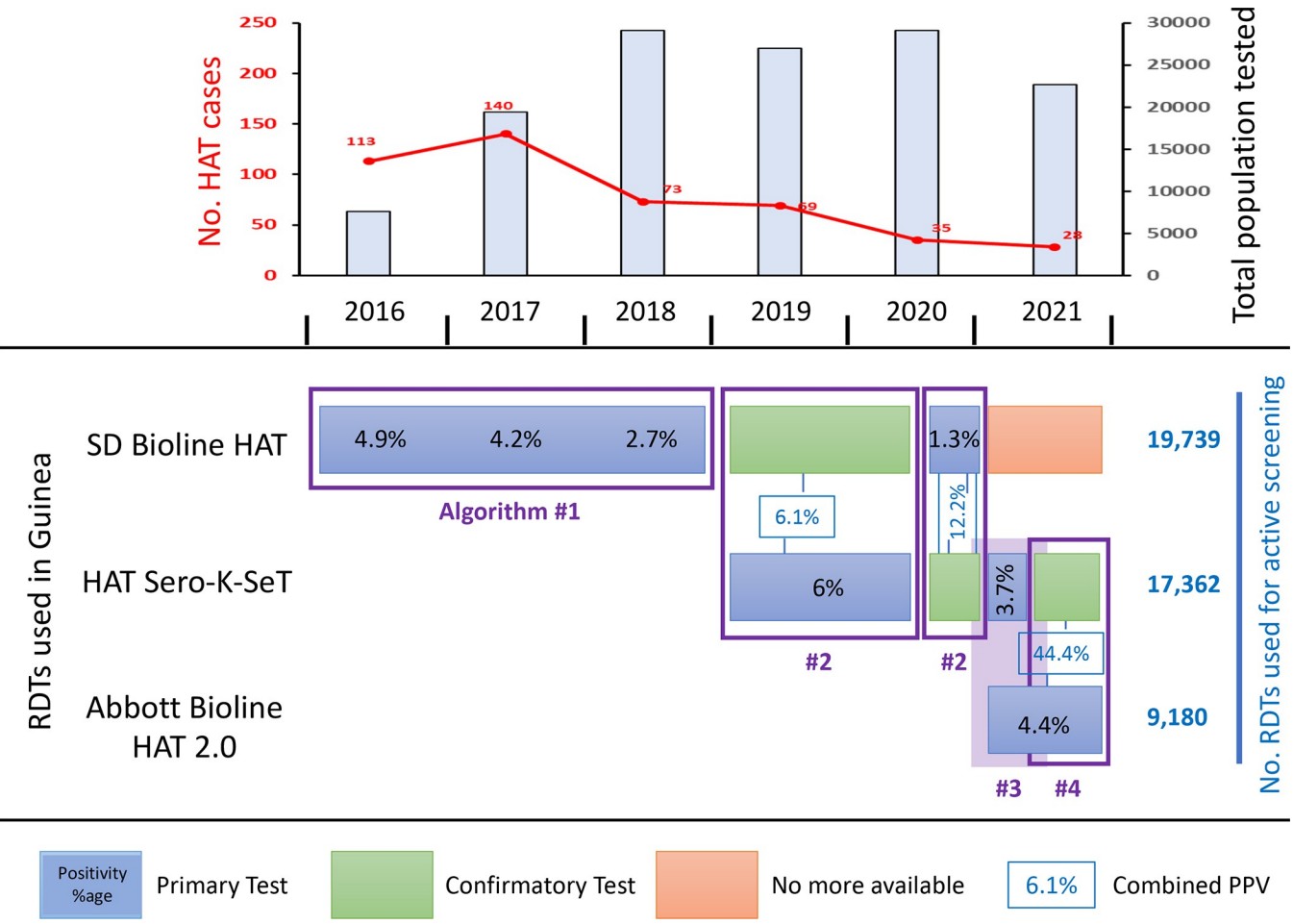

**Fig 1. Usage of the different HAT-RDTs for door-to-door active medical surveys from 2016 to 2021 in Guinea.** The evolution of the total annual number of HAT cases detected by both passive and active surveillance is shown on the upper graph. Different strategies have been successively implemented according to the availability and cost of the different RDTs. From 2016 to 2018, SD Bioline HAT was applied (details in Table 1). From 2019 to mid-2021, HAT Sero-K-SeT was used either alone, or in combination with SD Bioline HAT then with Abbott Bioline HAT 2.0 (details Table 2). From mid-2021 to date, Abbott Bioline HAT 2.0 was applied alone, then with HAT Sero-K-SeT used as secondary test (details in Table 3). The positivity rates (number of positive RDTs / number of screened individuals) were only calculated for RDTs used as first line tests. The Positive Predictive Values (PPVs) were calculated by combining results from primary and secondary RDTs.

(LLNNNN/N) in order to anonymize all the next steps and ensure confidentiality and privacy. A researcher or MD was in charge of this task during each campaign.

## HAT RDT procedure

Three HAT RDTs were used for the study: SD Bioline HAT, Abbott Bioline HAT 2.0 (Abbott Diagnostics Korea, Republic of Korea) and HAT Sero-K-SeT (Coris BioConcept, Belgium). SD Bioline HAT uses purified native *T. b. gambiense* variant surface glycoproteins (VSG) types LiTat 1.3 and 1.5 as antigens, visible in two separate test lines. HAT Sero-K-SeT also uses purified native VSG LiTat 1.3 and 1.5 as antigens but combined in a single test line consisting of a mix of both glycoproteins. The Abbott Bioline HAT 2.0 includes two recombinant antigens as two separate test lines, ISG65 and VSG LiTat 1.5 [6].

All tests were performed according to the manufacturers' recommendations. Briefly, SD Bioline HAT and Abbott Bioline HAT 2.0 tests were performed with 20 µl of blood dispensed

in the specimen well marked 'S'. Then, 4 drops of assay diluent were added by holding the assay diluent bottle vertically. For HAT Sero-K-SeT, 25 μl of blood were applied in the sample well marked '1', followed by two drops of BL-A buffer in the buffer well marked '2', after which the test device was re-inserted into its pouch to avoid reaction with light and limit dehydration. All results were observed 15 minutes after application of the buffer. In case the control line did not appear, the test result was considered invalid. The HAT Sero-K-SeT was positive if the test line was positive, the SD Bioline HAT and Abbott Bioline HAT 2.0 were considered positive if at least 1 test line was read as positive. Any positive test was assessed by a second reader for confirmation.

## HAT serological screening strategies during door-to-door campaigns

The active screening campaigns took place in the three active HAT foci (Boffa, Dubreka and Forecariah, in red in Fig 2), located along the Guinean Atlantic coast, between January 2016 and October 2021, and implemented by the HAT-NCP. The overall prevalence in these foci evolved from 1.33% in 2016 to 0.26% in 2020, with an incidence of 1 case per 10.000 inhabitants in 2020 (Fig 1). Before the start of each activity, a preparatory visit to the identified sites was conducted, followed by at least two-days of awareness raising activities among the

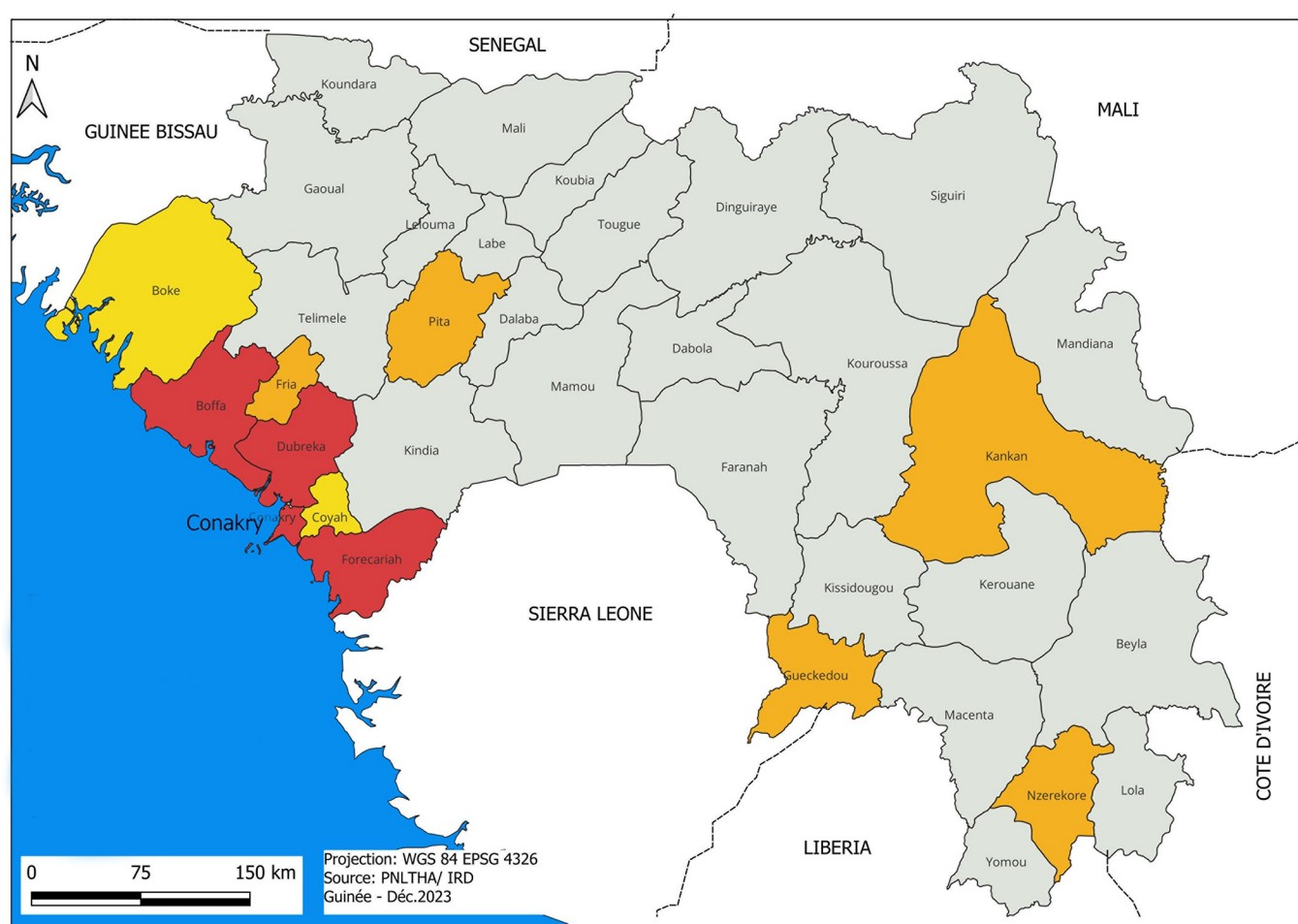

**Fig 2. Map of the HAT foci in Guinea.** Endemic transmission foci in red, old foci with last cases reported before 2004 in orange, zones at risk in yellow, Atlantic Ocean in blue. This map was elaborated in-house with QGIS 3.28.12 from an OSM standard layer (www.openstreetmap.org).

population, according to their availability that was mostly ruled by the tides for the islands and the agrarian calendar for the continent. Most villagers were carrying out seasonal activities such as fishing, agriculture, logging and salt extraction in these at-risk areas. The awareness raising was organized by the focal points of each household plus a community agent from the villages identified by focus groups. During the screening campaigns, participants were systematically selected for enrolment on the basis of their serological positivity by at least one test, without any other selection criteria. Then, individuals were only enrolled after having signed an informed consent.

After awareness raising activities aiming to inform and prepare the population, finger prick blood samples were collected with capillaries from volunteers, and applied to the RDTs (SD Bioline HAT in 2016–20, HAT Sero-K-SeT in 2019–21 and/or Abbott Bioline HAT 2.0 in 2021). The serological screening was performed by small mobile teams each composed of two health workers and one community health worker who were meeting and screening the population on a D2D basis. Serological suspects were referred to the parasitological confirmation laboratory where the direct examination of lymph node aspirates and/or the mini-anion exchange centrifugation test (mAECT) on buffy coats were performed [4].

According to the variable availability of the RDTs, different diagnosis algorithms have been elaborated (Fig 1).

- Algorithm 1: Only one test used for serological screening. Any individual testing positive was referred for parasitological confirmation. In this case, the positivity rate and PPV of the test was calculated.

- Algorithm 2: One RDT used as the primary screening test. Any individual testing positive to the primary test was then tested with the secondary test. Only individuals testing positive to the secondary test, or presenting specific clinical symptoms (cervical lymph nodes, severe pruritus, behavior disorders, weight loss or malaria resistant fevers), were referred for parasitological confirmation. In this case, the positivity rate of the primary test and the combined PPV (number of confirmed cases / numbers of double positive) were calculated.

- Algorithm 3: All the population was screened with two tests. Only double positive individuals or individuals testing positive to only one test but with specific clinical signs were referred for parasitological confirmation. In this case, we calculated the positivity rate of each RDT and the combined PPV of the two tests.

- Algorithm 4: One RDT was used as the primary screening test. Any individual testing positive to the primary test was referred for parasitological confirmation and tested for the secondary test. In this case, we calculated the positivity rate of the primary RDT, the PPV of the primary RDT and the combined PPV of the double positive subjects.

From 2016 to 2018, SD Bioline HAT was applied. From 2019 to mid-2021, HAT Sero-K-SeT was used either alone, or in combination first with SD Bioline HAT and then with Abbott Bioline HAT 2.0. From May to October 2021, Abbott Bioline HAT 2.0 was applied first, with HAT Sero-K-SeT used as secondary test.

## Retrospective analyses on samples from the HAT-NCP Biobank

To evaluate the sensitivity and specificity of the three RDTs, 189 plasma samples, including 67 from confirmed HAT patients and 122 from CATT-negative controls, were analyzed. We determined that the minimum required sample size allowing for detecting significant differences between the two groups with 95% confidence was 63 samples per group (hypothetical estimation of 20% and 50% positivity rates in the control and case groups respectively, 95%

power). They all originated from the HAT-NCP biobank collected from 2017 to 2020 during active screening campaigns. RDTs were performed according to manufacturers' instructions. Reading was performed by 2 independent readers. Without consensus, the opinion of a third reader was sought to validate the result.

### Assessment of putative cross-reaction events in *Plasmodium falciparum* infected participants

To exclude any putative cross-reaction events of Abbott Bioline HAT 2.0 with *Plasmodium (P.) falciparum* infection-related factors, the SD Bioline Malaria Ag P.f rapid test, detecting a *P. falciparum* histidine-rich protein II (HPR-II) antigen, was also used. First, in October and November 2021, in Boffa and Forecariah, SD Bioline Malaria Ag P.f RDT was systematically performed on 202 individuals in parallel to the Abbott Bioline HAT 2.0 in the context of a diagnostic integration pilot study launched by the HAT-NCP. Then, because of a suspicion of cross-reactions, samples from 479 individuals living in HAT-free areas where malaria is endemic were tested in the Friah health center in September 2021 (n = 144) and in the St-Gabriel dispensary of Matoto, Conakry, in October-November 2021 (n = 335). The test was performed according to the manufacturer indications. 5 μl of blood was taken with the disposable inverted cup provided and applied into the round specimen well. After that, 4 drops of assay diluent were vertically dispensed into the square assay diluent well. The result was read after 15 minutes.

### Statistical analysis

Positive Predictive Values (PPV) of individual test and combination of tests were calculated. Sensitivity and specificity were calculated on confirmed HAT cases versus CATT-negative control group. The positivity rates of SD HAT Bioline 2.0 and SD Bioline Malaria Ag P.f RDTs sub-groups were compared using Fisher exact tests. P-values lower than 0.05 were considered significant.

## Results and discussion

### First steps with RDTs in active screening: Performance of SD Bioline HAT

From 2016 to 2018, SD Bioline HAT was used in the three active Guinean transmission foci (Boffa, Dubreka and Forecariah) (Figs 1, 2 and Table 1). The absence of active screening during the Ebola outbreak led to an instant reduction in case detection. Consequently, the total number of new cases detected after this outbreak was relatively high (113 in 2016, and 140 in 2017) (Fig 1). Then, a sustained surveillance (2 active screening campaigns per year in each transmission focus) with a close case management, associated with the progressive implementation of vector control (since 2016 in Boffa West and Dubreka, and since 2018 in Forecariah), led to a rapid decrease in the total number of new cases detected down to 28 in 2021, reflecting an overall decrease of parasite transmission in all foci (Fig 1). This trend can also be assessed by the decrease in the positivity rate of SD Bioline HAT tests used for D2D active screening from 4.9% (n = 2,884) in 2016 to 1.3% (n = 4,814) in 2021, as well as by the decrease in the PPV of this RDT from 38.6% in 2016 to 8.8% in 2020, stabilized at 9.1% in 2021 (Table 1). Interestingly, the combination of the Sero-K-SeT as a second line RDT in case of SD Bioline HAT positivity at first did not improve the overall PPV of the diagnostic algorithm (no statistical difference in PPV between SD and SD+Sero-K-SeT) (Table 1).

**Table 1. Usage and performance of SD Bioline HAT for D2D active screening in Guinea from 2016 to 2020.**

| Primary test | Secondary test | Focus | Period | Nb of subjects screened | Nb of SD Bioline HAT + | Nb of Sero-K-SeT + | Nb of confirmed HAT cases | Referral rate | SD Bioline HAT positivity | PPV of SD Bioline HAT | PPV of SD Bioline HAT & Sero-K-SeT |
|---|---|---|---|---|---|---|---|---|---|---|---|
| SD Bioline HAT | - | Boffa | 2016 | 1 386 | 80 | - | 45 | 100% | 5.8% | 56.3% | - |
| SD Bioline HAT | - | Forecariah | 2016 | 1 498 | 60 | - | 9 | 100% | 4.0% | 15.0% | - |
| **SD Bioline HAT** | **-** | | **2016** | **2 884** | **140** | **-** | **54** | **100%** | **4.9%** | **38.6%** | **-** |
| SD Bioline HAT | - | Boffa | 2017 | 1 139 | 37 | - | 11 | 100% | 3.2% | 29.7% | - |
| SD Bioline HAT | - | Forecariah | 2017 | 1 412 | 69 | - | 25 | 100% | 4.9% | 36.2% | - |
| **SD Bioline HAT** | **-** | | **2017** | **2 551** | **106** | **-** | **36** | **100%** | **4.2%** | **34.0%** | **-** |
| SD Bioline HAT | - | Boffa | 2018 | 2 892 | 106 | - | 4 | 74% | 3.7% | 5.1% | - |
| SD Bioline HAT | - | Dubreka | 2018 | 4 239 | 56 | - | 4 | 79% | 1.3% | 9.1% | - |
| SD Bioline HAT | - | Forecariah | 2018 | 2 359 | 93 | - | 10 | 89% | 3.9% | 12.0% | - |
| **SD Bioline HAT** | **-** | | **2018** | **9 490** | **255** | **-** | **18** | **80%** | **2.7%** | **8.8%** | **-** |
| SD Bioline HAT | Sero-K-SeT * | Boffa | 2020 | 1 918 | 9 | 8 | 1 | 100% | 0.5% | 11.1% | 12.5% |
| SD Bioline HAT | Sero-K-SeT* | Forecariah | 2020 | 2 896 | 52 | 33 | 4 | 88% | 1.8% | 8.7% | 12.1% |
| **SD Bioline HAT** | **Sero-K-SeT *** | | **2020** | **4 814** | **61** | **41** | **5** | **90%** | **1.3%** | **9.1%** | **12.2%** |
| | | | | **19 739** | **562** | **-** | **113** | **-** | **2.8%** | **22.3%** | **-** |

\* If SD Bioline HAT + then Sero-K-SeT and parasitological exam = Algorithm 4.

PPV: Positive Predictive Value. The referral rate represents the proportion of serosuspects investigated in the lab for parasitological confirmation.

## Combining RDTs

In 2019 and 2020, the uncertainty regarding SD Bioline HAT availability became a major concern and it was decided to use HAT Sero-K-SeT as the primary test for D2D screening, while maintaining the SD Bioline HAT available for the passive surveillance only (Fig 1 and Table 2). HAT Sero-K-SeT was indeed the only alternative RDT available in 2019, but its PPV was found to be limited in the context of D2D active screening (5.6%, n = 5,406) (Table 2). The high number of positive suspects requiring confirmation by parasitological examination (mAECT) was overloading the field laboratory, with an increased waiting time for sero-suspects to be managed, hence with a substantial loss of individuals during referral (76% mean referral rate in 2019) (Table 2). The need for a confirmatory and / or complementary RDT to increase the PPV of our initial diagnostic algorithm prompted us to compare different algorithms (Table 2). Although no PPV improvement was observed when combining HAT Sero-K-SeT as primary RDT and SD Bioline HAT in second line (combined PPV of 6.1%, n = 9,997 in 2019–20), this diagnostic algorithm was at least able to limit the laboratory workload and the loss of serological suspects (92% mean referral rate) (Table 2).

**Table 2. Usage and performance of Coris HAT Sero-K-SeT for D2D active screening in Guinea from 2019 to 2021.**

| Primary test | Secondary test | Focus | Period | Nb of subjects screened | Nb of Sero-K-SeT + | Nb of SD Bioline HAT + | Nb of Abbott Bioline HAT 2.0 + | Nb of confirmed HAT cases | Referral rate | Sero-K-SeT positivity | PPV of Sero-K-SeT | PPV of Sero-K-SeT & SD Bioline HAT | PPV of Sero-K-SeT & Abbott Bioline HAT 2.0 |
|---|---|---|---|---|---|---|---|---|---|---|---|---|---|
| Sero-K-SeT | - | Boffa | Jun-19 | 2 924 | 156 | - | - | 9 | 96% | 5.3% | 6.0% | - | - |
| Sero-K-SeT | - | Dubreka | Jun-19 | 2 482 | 146 | - | - | 4 | 55% | 5.9% | 4.9% | - | - |
| **Sero-K-SeT** | **-** | | | **5 406** | **302** | | | **13** | **76%** | **5.6%** | **5.6%** | **-** | **-** |
| Sero-K-SeT | SD Bioline HAT * | Forecariah | Nov-19 | 2 477 | 88 | 23 | - | 1 | 100% | 3.6% | - | 4.3% | - |
| Sero-K-SeT | SD Bioline HAT * | Dubreka | Jan-20 | 2 310 | 156 | 46 | - | 0 | 100% | 6.8% | - | 0.0% | - |
| Sero-K-SeT | SD Bioline HAT * | Boffa | Sep-20 | 3 569 | 133 | 26 | - | 1 | 100% | 3.7% | - | 3.8% | - |
| Sero-K-SeT | SD Bioline HAT * | Forecariah | Sep-20 | 2 869 | 270 | 63 | - | 6 | 98% | 9.4% | - | 9.7% | - |
| Sero-K-SeT | SD Bioline HAT * | Forecariah | Nov-20 | 3 559 | 264 | 70 | - | 1 | 84% | 7.4% | - | 1.7% | - |
| **Sero-K-SeT** | **SD Bioline HAT *** | | | **9 997** | **667** | **159** | | **9** | **92%** | **6.7%** | **-** | **6.1%** | **-** |
| Sero-K-SeT | Abbott Bioline HAT 2.0 ** | Dubreka | Fev-21 | 1 959 | 73 | - | 37 | 0 | 84% | 3.7% | - | - | 0.0% |
| **Sero-K-SeT** | **Abbott Bioline HAT 2.0 **** | | | **1 959** | **73** | **-** | **37** | **0** | **84%** | **3.7%** | **-** | **-** | **0.0%** |
| | | | | 17 362 | 1042 | - | - | 22 | - | 6.0% | - | - | - |

* If Sero-K-SeT+ and SD Bioline HAT + then parasitological exam = Algorithm 3.

** If Sero-K-SeT+ and Abbott Bioline HAT 2.0 + then parasitological exam = Algorithm 3.

PPV: Positive Predictive Value. The referral rate represents the proportion of serosuspects investigated in the lab for parasitological confirmation.

## Second generation RDTs: Performances of Abbott Bioline HAT 2.0

In 2020, the SD Bioline HAT production stopped, yet Abbott Bioline HAT 2.0 was finally commercialized in 2021. Considering its low price, as compared to HAT Sero-K-SeT, it was tempting to push this test as the primary RDT for the D2D (Fig 1). Nevertheless, because no comparative study was available to assess its performance in West Africa, Abbott Bioline HAT 2.0 was immediately tested in the Boffa and Forecariah transmission foci to evaluate its PPV (Fig 1 and Table 3). A parallel assessment showed a higher positivity rate of Abbott Bioline HAT 2.0 (6.0%, n = 2,250) as compared to HAT Sero-K-SeT (1.9%), with a combined PPV of 20.0% (Table 3). However, an evaluation of Abbott Bioline HAT 2.0 alone revealed a low PPV of 3.9% (n = 6,930) (Table 3). This was improved by using Abbott Bioline HAT 2.0 in first line and HAT Sero-K-SeT as a secondary test before confirmation, with a combined PPV reaching 44.4% (Table 3), a statistically significant performance increase in favor of maintaining this latter diagnostic algorithm (p = $3.1 \times 10^{-6}$ by Fisher; OR = 18.9 [5.1–72.9]). Interestingly, a significant increase of the PPV was observed when combining Abbott Bioline HAT 2.0 with HAT Sero-K-SeT, whereas no significant increase was observed for HAT Sero-K-SeT and SD Bioline HAT. This is likely because HAT Sero-K-SeT and SD Bioline HAT are both first generation RDTs using the same native antigens. In contrast, Abbott Bioline HAT 2.0 is a second

**Table 3. Usage and performance of Abbott Bioline HAT 2.0 for D2D active screening in Guinea in 2021.**

| Primary test | Secondary test | Focus | Period | Nb of subjects screened | Nb of Abbott Bioline HAT 2.0 + | Nb of Sero-K-SeT + | Nb of Abbott Bioline HAT 2.0 + and Sero-K-SeT + | Nb of confirmed HAT cases | Referral rate | Abbott Bioline HAT 2.0 positivity | Sero-K-SeT positivity | PPV of Abbott Bioline HAT 2.0 | PPV of Abbott Bioline HAT 2.0 & Sero-K-SeT |
|---|---|---|---|---|---|---|---|---|---|---|---|---|---|
| Abbott Bioline HAT 2.0 & Sero-K-SeT * | - | Forecariah | May-21 | 2 250 | 134 | 42 | 15 | 3 | 100% | 6.0% | 1.9% | - | 20.0% |
| **Abbott Bioline HAT 2.0 & Sero-K-SeT *** | **-** | | | **2 250** | **134** | **42** | **15** | **3** | **100%** | **6.0%** | **1.9%** | **-** | **20.0%** |
| Abbott Bioline HAT 2.0 | Sero-K-SeT ** | Boffa | Oct-21 | 3 523 | 147 | 11 | 11 | 3 | 67% | 4.2% | - | 3.0% | 27.3% |
| Abbott Bioline HAT 2.0 | Sero-K-SeT ** | Forecariah | Oct-21 | 3 407 | 125 | 7 | 7 | 5 | 84% | 3.7% | - | 4.8% | 71.4% |
| **Abbott Bioline HAT 2.0** | **Sero-K-SeT **** | | | **6 930** | **272** | **18** | **18** | **8** | **75%** | **3.9%** | **-** | **3.9%** | **44.4%** |
| | | | | **9 180** | **406** | **-** | **33** | **11** | **-** | **4.4%** | **-** | **-** | **33.3%** |

* If Abbott Bioline HAT 2.0 + and Sero-K-SeT + then parasitological exam = Algorithm 3.

** If Abbott Bioline HAT 2.0 + then Sero-K-SeT and parasitological exam = Algorithm 4.

PPV: Positive Predictive Value. The referral rate represents the proportion of serosuspects investigated in the lab for parasitological confirmation.

generation RDT using recombinant antigens, including one that was not included in the first-generation RDTs. Hence, maintaining the production of first generation RDTs would be important, especially in the perspective of conducing active screening campaigns.

The only prospective case-control evaluation of Abbott Bioline HAT 2.0 available in the literature is that of Lumbala *et al.* [6]. The Abbott Bioline HAT 2.0 used for active screening in the studied DRC foci showed a specificity of 99.1% and a sensitivity of 54.8%. Here, Abbott Bioline HAT 2.0 was compared to the previous RDTs used in Guinea, based on a combination of fragmented retrospective observations and prospective evaluations collected over 6 years of medical surveys on more than 46,000 individuals.

It is noteworthy that 3 parasitologicaly confirmed cases initially identified by Abbott Bioline HAT 2.0 were negative to HAT Sero-K-SeT. Also, a much higher number of non-confirmed subjects tested positive to Abbott Bioline HAT 2.0 as compared to Sero-K-SeT (Table 3). At this step, a confirmation of the low PPV of Abbott Bioline HAT 2.0 observed in the context of D2D active screening was required. However, due to the COVID-19 pandemic restrictions and the unstable political context in Guinea in the second half of 2021, implementing another state-of-the art RDT comparison by systematic testing of all subjects with different RDTs revealed to be challenging for the HAT-NCP.

## Retrospective comparison of RDTs

Therefore, a retrospective evaluation of all 3 RDTs was conducted on 189 plasma samples from the HAT-NCP biobank frozen between 2017 and 2020 (Table 4). This retrospective

**Table 4. Comparison of the sensitivity and specificity of HAT RDTs on frozen plasma samples from the HAT-NCP biobank (2017–2020) as compared to the initial parasitological diagnostic by mAECT and/or direct examination of lymph node aspirates.**

| % [95% CI] | HAT Sero-K-SeT | SD Bioline HAT | Abbott Bioline HAT 2.0 |
|---|---|---|---|
| Sensitivity (Cases, n = 67) | 88.1% [78.2–93.8%] | 64.2% [52.2–74.6%] | 94.0% [85.6–97.7%] |
| Specificity (Control, n = 122) | 98.4% [94.2–99.5%] | 98.4% [94.2–99.5%] | 83.6% [76.0–89.1%] |

analysis confirmed the higher sensitivity and the lower specificity of Abbott Bioline HAT 2.0 compared to SD Bioline HAT and HAT Sero-K-SeT (Table 4). The sensitivity of HAT Sero-K-SeT and SD Bioline HAT observed in this study was lower than that estimated from the DiTECT-HAT clinical trial where HAT Sero-K-SeT and SD Bioline HAT displayed sensitivities of 100% and 93.8%, respectively, in passive detection [8]. This difference can be explained by at least two points: (1) all samples for the retrospective study were collected during active screening campaigns from patients with limited clinical signs, and (2) frozen plasma may not be optimal for RDT testing. Nevertheless, it allowed a direct comparison between the three RDTs used in this analysis.

## HAT RDTs and malaria

During the D2D screening campaigns organized by the HAT-NCP in Boffa and Forecariah in October and November 2021, respectively, a malaria-RDT (SD Bioline Malaria Ag P.f) was also performed in parallel to the Abbott Bioline HAT 2.0 in the context of a diagnostic integration pilot study. Among 202 subjects tested positive with Abbott Bioline HAT 2.0, 84 were also positive with SD Bioline Malaria Ag P.f rapid test (41.6%). Although this observation was *a priori* not surprising due to the high prevalence of malaria in Guinea during the rainy season (May-November), we hypothesized that this high rate of double-positive tests could also result, at least partially, from possible interferences between malaria-related biological factors and Abbott Bioline HAT 2.0. Therefore, additional assessments of Abbott Bioline HAT 2.0 and SD Bioline Malaria Ag P.f rapid test positivity rates were performed by testing 479 subjects living in two distinct HAT-free malaria-endemic areas with both RDTs (Table 5). All subjects tested positive with Abbott Bioline HAT 2.0 were negative with mAECT, but a significantly higher proportion of subjects positive with Abbott Bioline HAT 2.0 were also positive with SD Bioline Malaria Ag P.f rapid test (Table 5). These results suggest a possible cross-reaction of Abbott Bioline HAT 2.0 with malaria-related biological factors in about 10% of malaria cases. Lumbala et al. [9] previously evaluated an RDT prototype for combined diagnosis of malaria and sleeping sickness. However, no details on the positivity of Abbott Bioline HAT 2.0 alone in malaria-positive vs. malaria-negative subjects were provided.

## RDTs in active screening vs. passive detection

The positivity rate of a RDT is expected to be lower in active screening than in passive surveillance. For instance, the PPV of HAT Sero-K-SeT and SD Bioline HAT in active screening were

**Table 5. False positive results with Abbott Bioline HAT 2.0 in malaria-RDT negative vs. positive subjects in Friah (Local Health Center, September 2021) and Conakry (Saint Gabriel Dispensary, October-November 2021).**

| HAT free regions | Abbott Bioline HAT 2.0 positivity in malaria-RDT negative | Abbott Bioline HAT 2.0 positivity in malaria-RDT positive | p value |
|---|---|---|---|
| Friah | 0/112 (0%) | 5/32 (15.6%) | 0.0004 |
| Conakry | 1/264 (0.4%) | 4/71 (5.6%) | 0.008 |
| Combined | 1/376 (0.3%) | 9/103 (8.7%) | $6.10^{-6}$ |

5.6% and 22.3%, respectively, whereas the PPV of these tests in passive detection were 45.2% and 48.4%, respectively [8]. This is mostly because mostly people presenting specific clinical signs are tested for HAT in peripheral health centers. However, a surprising observation was the limited differences in PPV observed for Abbott Bioline HAT 2.0 with positivity rates of 1.1% (132 / 11,496) and 4.4% (406 / 9,180) in passive and active detection, respectively (HAT-NCP annual report 2021). Several hypotheses could explain this observation: (1) the external test conditions (humidity and temperature inside versus outside a building), and/or the reading sensitivity with low intensity bands could impact the RDT output [10]; (2) people consulting with fever in peripheral health centers are first tested for malaria, and malaria-positive patients are not necessarily further tested for HAT, hence the possible cross-reactivity of Abbott Bioline HAT 2.0 with malaria-related factors would be limited in the passive detection system.

### RDTs algorithm

The low RDT PPVs observed in active screening could likely be improved by using optimal serial RDT algorithms also considering key rational implementation parameters such as test availability, RDT price, overall RDT-dependent parasitological confirmation algorithm costs and the associated human resources. Based on the positivity rates of Abbott Bioline HAT 2.0 alone and double positives in Abbott Bioline HAT 2.0 and HAT Sero-K-SeT observed in this study (Table 3), we compared the cost-effectiveness of two algorithms with Abbott Bioline HAT 2.0 being either used alone, or as a primary screening test associated to a HAT Sero-

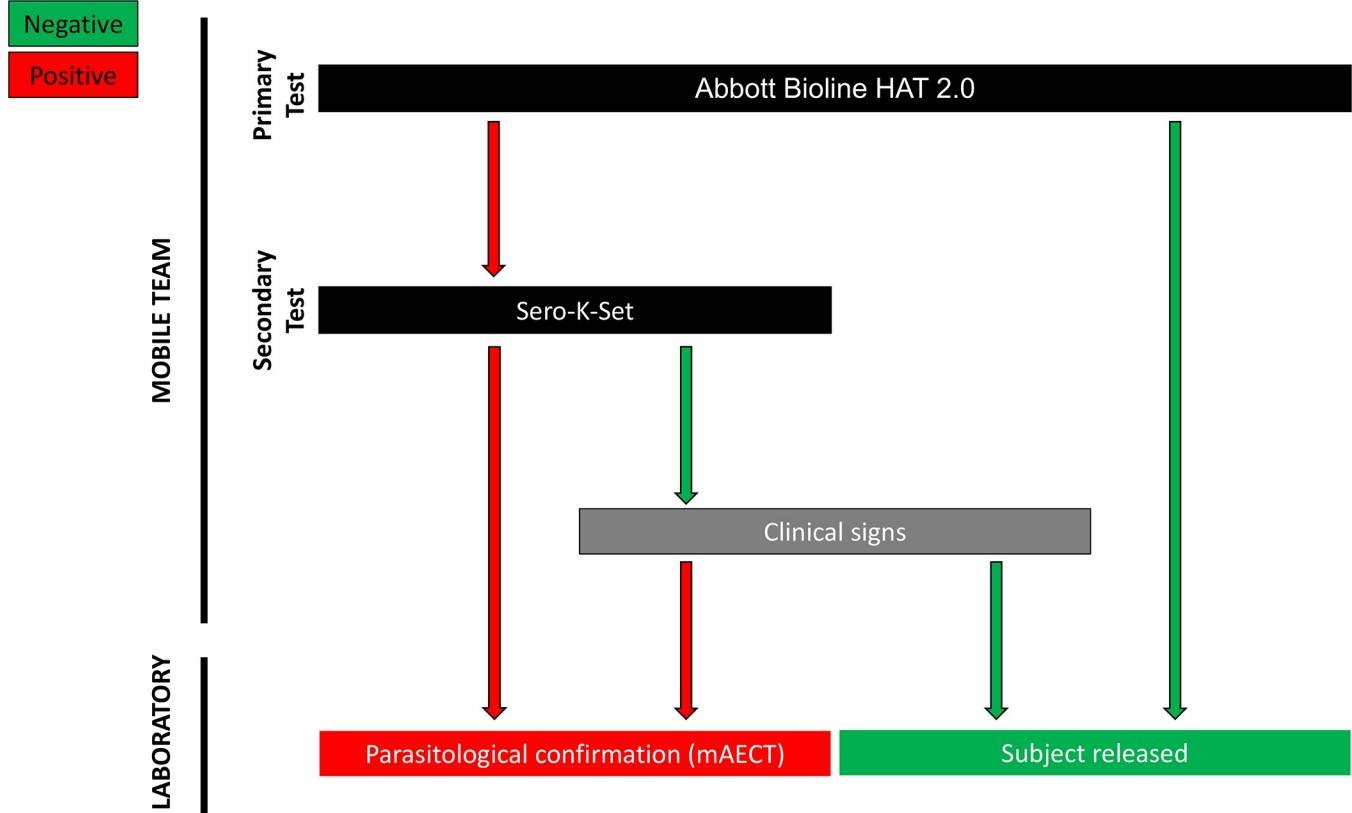

**Fig 3. Current algorithm used by the HAT-NCP to detect HAT cases by active screening in Guinea.** mAECT: mini-anion exchange centrifugation test.

K-SeT secondary test. The combination algorithm enables a significant decrease in both the human resource costs required for parasitological confirmation (divided by 2), and the overall transport costs and ecological impact (distance divided by more than 12), with a modest impact on the cost of diagnostic supplies (i.e., mostly mAECT costs).

## Conclusion

Despite the low specificities of RDTs in active screening, their optimal combination allows reaching improved PPVs for HAT diagnosis. The PPV was also improved by considering specific clinical signs in case of negative secondary tests. Based on all these recent (2016–2021) observations and comparisons, the resulting and currently used active screening algorithm implemented in Guinea is presented in Fig 3. However, further studies will be needed to determine whether a significant number of HAT cases are missed using these new algorithms. A state-of-the-art prospective comparative study is currently ongoing in Guinea and Côte d'Ivoire for comparing all currently available HAT RDTs and new HAT RDT prototypes for D2D active screening.

## Acknowledgments

We warmly thank all the team of the Programme National de Lutte contre la Trypanosomiase Humaine Africaine in Guinea, as well as all collaborators of the Boffa, Forecariah and Dubreka Health Districts. We thank Dr. Moïse Kagbadouno for his help with the map drawing. We are also grateful to the medical team led by Paul Bioche at the Saint Gabriel Dispensary in Conakry.

## Author Contributions

**Conceptualization:** Mamadou Camara, Jean-Mathieu Bart, Brice Rotureau, Bruno Bucheton.

**Data curation:** Oumou Camara, Justin Windingoudi Kaboré, Mamadou Leno, Jean-Mathieu Bart, Brice Rotureau, Bruno Bucheton.

**Formal analysis:** Oumou Camara, Justin Windingoudi Kaboré, Jean-Mathieu Bart, Brice Rotureau, Bruno Bucheton.

**Funding acquisition:** Mamadou Camara, Brice Rotureau, Bruno Bucheton.

**Investigation:** Oumou Camara, Justin Windingoudi Kaboré, Aïssata Soumah, Mamadou Leno, Dominique N'Diaye, Jean-Mathieu Bart, Brice Rotureau, Bruno Bucheton.

**Methodology:** Oumou Camara, Justin Windingoudi Kaboré, Mamadou Leno, Mamadou Camara, Jean-Mathieu Bart, Brice Rotureau, Bruno Bucheton.

**Project administration:** Mohamed Sam Bangoura, Adrien Marie Gaston Belem, Mamadou Camara, Jean-Mathieu Bart, Brice Rotureau, Bruno Bucheton.

**Supervision:** Adrien Marie Gaston Belem, Mamadou Camara, Jean-Mathieu Bart, Brice Rotureau, Bruno Bucheton.

**Validation:** Mamadou Camara, Jean-Mathieu Bart, Brice Rotureau, Bruno Bucheton.

**Visualization:** Jean-Mathieu Bart, Brice Rotureau, Bruno Bucheton.

**Writing – original draft:** Jean-Mathieu Bart, Brice Rotureau, Bruno Bucheton.

**Writing – review & editing:** Oumou Camara, Sylvain Biéler, Jean-Mathieu Bart, Brice Rotureau, Bruno Bucheton.

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
