## [Decision Letter · Decision Letter 0]

2 Oct 2023

Dear Dr Rotureau,

Thank you very much for submitting your manuscript "Conducting active screening for human African trypanosomiasis with rapid diagnostic tests: the Guinean experience (2016-2021)" for consideration at PLOS Neglected Tropical Diseases. As with all papers reviewed by the journal, your manuscript was reviewed by members of the editorial board and by several independent reviewers. In light of the reviews (below this email), we would like to invite the resubmission of a significantly-revised version that takes into account the reviewers' comments. 

The manuscript has been reviewed by 4 reviewers and all of them requested multiple revisions (see below). 

Two reviewers notified concerns about ethical approval and informed consent.

This point is important and needs to be addressed properly.

We cannot make any decision about publication until we have seen the revised manuscript and your response to the reviewers' comments. Your revised manuscript is also likely to be sent to reviewers for further evaluation.

Sincerely,

Jaap J van Hellemond

Academic Editor

Ricardo Fujiwara

Section Editor

The manuscript has been reviewed by 4 reviewers and all of them requested multiple revisions (see below). 

Two reviewers notified concerns about ethical approval and informed consent.

This point is important and needs to be addressed properly.

Reviewer's Responses to Questions

**Key Review Criteria Required for Acceptance?**

**Methods**

-Are the objectives of the study clearly articulated with a clear testable hypothesis stated?

-Is the study design appropriate to address the stated objectives?

-Is the population clearly described and appropriate for the hypothesis being tested?

-Is the sample size sufficient to ensure adequate power to address the hypothesis being tested?

-Were correct statistical analysis used to support conclusions?

-Are there concerns about ethical or regulatory requirements being met?

Reviewer #1: see below

Reviewer #2: Include additional information on the study site regarding the prevalence and incidence of HAT so that your readers can understand the problem better.

How was the population sensitization done before sample collection? 

Did you confirm that the sensitization was effective? 

How was the sampling/selection of volunteers done? 

How did you avoid selection bias?

On putative cross-reactions in P. falciparum infected participants, did you control for potential hrp2/3 gene deletion?

Ethical Considerations

It is unclear if ethical clearance was sought before the medical surveillance campaigns. It would be good to include the name of the Institutional Research/Ethics Committee/Board. How did you ensure the confidentiality and privacy of the participants?

Was ethical approval sought before accessing the samples in the HAT-NCP biobank?

How did you determine the sample size for the HAT-NCP biobank samples?

In the various algorithms, how did you ensure the principle of beneficence, especially in participants whose screening tests turned negative and were asymptomatic or had atypical symptoms?

Reviewer #3: yes, the objectives of the study are clear, however, the manuscript lacks a bit of structure, some parts are very confusing for the reader and some descriptions need more specific details.

Reviewer #4: The materials and methods are clearly described and appropriate to the study, with a focus on assessing the sensitivity and specificity of the respective RDTs in active HAT foci following a change in the availability of the SD bioline HAT RDT. The study procedures are clearly described and the use of a retrospective analysis provides the opportunity to directly compare the tests using samples whose infection status has already been determined. Ethical considerations are clearly stated and were approved by the appropriate national bodies.

**Results**

-Does the analysis presented match the analysis plan?

-Are the results clearly and completely presented?

-Are the figures (Tables, Images) of sufficient quality for clarity?

Reviewer #1: see below

Reviewer #2: The findings are really interesting and informative.

I’m not sure why you mixed the results and discussion, but you seem to have lost some clarity. Perhaps you could have presented the results first, followed by a discussion bit, either for every key result or totally separate. For instance, the first paragraph in this section starts with statements that sound like you are explaining something that should have been in the background or an explanation for a key finding.

Reviewer #3: The results should be re-structured a bit, double check figures and tables if they are correctly numbered and refer to the correct tables and figures.

Reviewer #4: The authors present a clear summary of the results, broken down into appropriate sections. The retrospective analysis comparing the three RDTs and cross reactivity survey are of particular interest as they highlight the key results and challenges faced during screening in a malaria endemic region. The identification of possible cross reactivity with malaria antigens warrants further investigation, especially as HAT cases continue to decline in prevalence.

The figures are basic but clear and appropriate to the results. The tables, which present the core results, would benefit from some revision to improve legibility as the included in line versions are far too small. The authors may wish to consider moving some details to the supplementary section and focus on only a small number of the currently included columns in the main text. With the exception of Figure 1 there are limited figure/table legends and more details are required here.

**Conclusions**

-Are the conclusions supported by the data presented?

-Are the limitations of analysis clearly described?

-Do the authors discuss how these data can be helpful to advance our understanding of the topic under study?

-Is public health relevance addressed?

Reviewer #1: see below

Reviewer #2: The conclusion and recommendations are relevant. The study limitations are not clearly stated.

Reviewer #3: yes

Reviewer #4: The conclusions drawn throughout the combined results/discussion are appropriate and clearly presented. The authors discuss potential reasons for the differences observed between the RDTs and the differences observed between this sample and the higher sensitivities observed in the DiTECT-HAT clinical trial. The public health relevance is clearly discussed throughout, from both the perspective of sensitivity/specificity and resource costs that can be saved by using the combination algorithm described in the manuscript. They highlight an in progress study that will extend these results to both currently used RDTs and new prototypes designed for D2D screening.

**Editorial and Data Presentation Modifications?**

Reviewer #1: see below

Reviewer #2: (No Response)

Reviewer #3: Minor revision

Reviewer #4: The study is clearly written throughout and overall well presented. As already noted some of tables are too dense and will require reformatting to use either a full page design or a simplification to enable the most important details to be clearly visible. This revision is should not be hard to make.

**Summary and General Comments**

Reviewer #1: This manuscript is addressing an important issue in neglected tropical diseases, i.e. the correct diagnosis of HAT. The manuscript is of interest for the public of this journal. In principle the research is sufficiently designed and presented. There are however issues that require the attention of the authors and that need to be resolved.

The use of English requires some attention; there are some minor errors found within the manuscript. Possibly this can be done through an editorial check.

An important issue is the ethical consideration. There seems to be no appropriate review and approval by competent ethical review bodies in the country concerned. Furthermore, although an informed consent form has been signed by participants, probably at the time of signature they were not informed about the current study. This is an issue and must be resolved.

The methodology section is rather detailed and could be shortened. For example, lines 195-202 are much repeating lines 142-153 (although the first addressed the case of no consensus).

If possible batch numbers and expiration dates of tests used should be provided. 

Were any positive/negative control samples used during the campaigns to ensure proper use of the RDTs?

The retrospective analyses in samples from the Biobank raise some questions. It seems that plasma (or sera?) have been used from (non) cases; this needs to be clarified. Use of blood seems to be the preferred diagnostic sample (see lines 143 – 147). Could the use of plasma/sera affect the test performance? Samples of CATT negative controls were used. Did these comprise other endemic diseases, like malaria, as this seems to be a concern. 

In the section statistical analysis it is mentioned that sensitivity/specificity were calculated on confirmed HAT cases. How were these cases confirmed? By parasitology? Or was it only serology (in that case, I have some doubts about the accuracy).

The tables must be modified to ensure unity of style. Currently many different lay-outs (and often hard to read) are being used. Headings to these tables should be more descriptive.

The so-called cost-effectiveness should be removed from the manuscript. CE calculations require a much more in depth study design and the current work is just a very rough description.

For clarity reasons, I would propose to separate results and discussion into two separate sections.

Reviewer #2: The study topic is helpful. However, the authors need to improve the methods section to assure readers that ethical principles were satisfactorily considered.

Reviewer #3: dear Authors, 

The manuscript describes an interesting topic that is gaining more importance for the diagnosis of HAT. 

There is an overall lack of structure in the way the results are decribed and presented in the manuscript, making it difficult to understand for the reader. 

Detailed comments can be found in the attachement

Reviewer #4: The study by Camara et al aims to compare the sensitivity and specificity of the RDTs that have been used within Guinea to identify HAT infections as the country moves towards elimination trypanosomiasis as a public health concern. The study combines current datasets with a retrospective analysis in an appropriate manner that allows for a direct comparison of the three RDTs being studied. The study is relatively simple in design and not particularly novel but addresses an important question that has clear public health and policy implications as countries push towards the elimination of HAT, hence my recommendation that it is accepted with only minor revisions.

PLOS authors have the option to publish the peer review history of their article (what does this mean?). If published, this will include your full peer review and any attached files.

Reviewer #1: No

Reviewer #2: No

Reviewer #3: No

Reviewer #4: No
---

## [Decision Letter · Decision Letter 1]

15 Jan 2024

Dear Dr Rotureau,

Thank you very much for submitting your manuscript "Conducting active screening for human African trypanosomiasis with rapid diagnostic tests: the Guinean experience (2016-2021)" for consideration at PLOS Neglected Tropical Diseases. As with all papers reviewed by the journal, your manuscript was reviewed by members of the editorial board and by several independent reviewers. The reviewers appreciated the attention to an important topic. Based on the reviews, we are likely to accept this manuscript for publication, providing that you modify the manuscript according to the review recommendations. 

No further comments in addition to those of the reviewers.

Sincerely,

Jaap J van Hellemond

Academic Editor

Ricardo Fujiwara

Section Editor

No further comments in addition to those of the reviewers.

Reviewer's Responses to Questions

**Key Review Criteria Required for Acceptance?**

**Methods**

-Are the objectives of the study clearly articulated with a clear testable hypothesis stated?

-Is the study design appropriate to address the stated objectives?

-Is the population clearly described and appropriate for the hypothesis being tested?

-Is the sample size sufficient to ensure adequate power to address the hypothesis being tested?

-Were correct statistical analysis used to support conclusions?

-Are there concerns about ethical or regulatory requirements being met?

Reviewer #1: Ýes it is adequate

Reviewer #3: Methods section has significantly improved, the descriptions are more clear. 

The figure of the map is OK, but I think the quality should be improved, it is not possible to read the names of the provinces/foci, the font is blurry. 

It is not clear what when data collection for your analysis (study) ended, line 177 mentions january 2016 till May 2021 while line 222 talks about mid-2021 to date. pleas clarify if data collection ended may 2021 or december 2021? As I am assuming that screening campaigns continue to this date. 

The last paragraph of your methods sections refers to tables, in which the results are already presented. to avoid confusion, I suggest not to refer to the tables yet, only when describing the results. 

Line 232-237, you can finish your sentece by "RDTs were performed according manufactures instructions" you already explained briefly previously in lines 164-173, no need for repetition.

Lastly, in the results+discussion, you mention (434-437) analysis of costs (HR, transport, supplies...) however, it was not described in the methods how you determined these costs, where you collected and how you analysed.

Reviewer #4: (No Response)

**Results**

-Does the analysis presented match the analysis plan?

-Are the results clearly and completely presented?

-Are the figures (Tables, Images) of sufficient quality for clarity?

Reviewer #1: Ýes it is adequate

Reviewer #3: Combining Results and discussion is acceptable, in this case it did improve readability of the manuscript. 

Line 341 is very confusing. Abbot 2.0 was only available from mid 2021 onwards, so its impossible to evaluate its performance over the 6 year period. If I look at the tables and figures, Abbot was used in prospective observations for 6 months and also on all retrospective samples. But indeed the other RDTs were used during this 6 year period, so I'm assuming that you compared the performance of Abbot 2.0 (6 months) against other RDTs (6 years), am I correct? please clarify. 

Paragraph starting line 410, indeed surprising that higher positivity rate was found in active screening. there was no hypothesis given for this? Line 415, your sentence starts with Similarly, however, the result is quite the opposite: positivity rate higher in active screening while PPV higher in passive screening? Did I interpret you text right? 

However, higher PPV in passive screening is to be expected, since people are presenting with specific clinical signs. 

Therefore your hypothesis nr 3 should come in first place. Also hypothesis nr 1, I could not understand the reasoning behind this explanation. Can you please clarify how malaria-positive patients are not further tested for HAT influence the PPV? 

As mentioned before, the results of the cost analysis (line 434) are not shown in a table or figure, or in supplementary info, but also was not described in methods how data collection was done and how analysis was done.

Reviewer #4: (No Response)

**Conclusions**

-Are the conclusions supported by the data presented?

-Are the limitations of analysis clearly described?

-Do the authors discuss how these data can be helpful to advance our understanding of the topic under study?

-Is public health relevance addressed?

Reviewer #1: Ýes it is adequate

Reviewer #3: The final conclusion is acceptable, it would also have been noteworthy to mention what the best screening algorithm would be in passive screening.

Reviewer #4: (No Response)

**Editorial and Data Presentation Modifications?**

Reviewer #1: Ýes it is adequate, although I still prefer to see results and discussion seperate, but leave this to the editor

Reviewer #3: (No Response)

Reviewer #4: (No Response)

**Summary and General Comments**

Reviewer #1: Ýes it is adequate

Reviewer #3: The manuscript has significantly improved, however, there are minor issues that still need to be adressed before acceptance.

Reviewer #4: The authors have provided a thorough response to the comments by myself and the other reviews, making substantial changes to the manuscript to include additional details (especially with regards to ethical approval) and clarify the presentation. The table and figures are much improved and far more legible than was provided in the original submission.

At this point I am happy to recommend the manuscript for publication as while the work is not groundbreaking it is nonetheless valuable as the data will guide public health policy going forward and directly aid in the elimination of trypanosomiasis in the future.

PLOS authors have the option to publish the peer review history of their article (what does this mean?). If published, this will include your full peer review and any attached files.

Reviewer #1: No

Reviewer #3: No

Reviewer #4: No

Figure Files:

Data Requirements:

Reproducibility:

References

---

## [Editor Report · Decision Letter 2]

9 Feb 2024

Dear Dr Rotureau,

We are pleased to inform you that your manuscript 'Conducting active screening for human African trypanosomiasis with rapid diagnostic tests: the Guinean experience (2016-2021)' has been provisionally accepted for publication in PLOS Neglected Tropical Diseases.

Best regards,

Claudia Ida Brodskyn

Section Editor

Ricardo Fujiwara

Section Editor

---

## [Editor Report · Acceptance letter]

15 Feb 2024

Dear Dr Rotureau,

We are delighted to inform you that your manuscript, "Conducting active screening for human African trypanosomiasis with rapid diagnostic tests: the Guinean experience (2016-2021)," has been formally accepted for publication in PLOS Neglected Tropical Diseases.

Best regards,

Shaden Kamhawi

co-Editor-in-Chief

Paul Brindley

co-Editor-in-Chief
